# Synthesis and Biological Activities of C1-Substituted Acylhydrazone *β*-Carboline Analogues as Antifungal Candidates

**DOI:** 10.3390/molecules29153569

**Published:** 2024-07-29

**Authors:** Yujie Xu, Lishan Li, Jinghan Zhang, Yu Lan, Na Li, Junru Wang

**Affiliations:** 1College of Chemistry and Pharmacy, Northwest A&F University, Yangling 712100, China; xuyj91@nwafu.edu.cn (Y.X.); 2021051625@nwafu.edu.cn (L.L.); zhangjinghan0906@163.com (J.Z.); lanyu9920@nwafu.edu.cn (Y.L.); 2Department of Scientific Research Services, Sanya Yazhou Bay Center for Innocation and Development Co., Ltd., Sanya 572000, China

**Keywords:** *β*-carboline, acylhydrazone, antifungal activity, structure–activity relationship, cytotoxicity

## Abstract

In our ongoing work to create potential antifungal agents, we synthesized and tested a group of C1-substituted acylhydrazone *β*-carboline analogues **9a**–**o** and **10a**–**o** for their effectiveness against *Valsa mali*, *Fusarium solani*, *Fusarium oxysporum*, and *Fusarium graminearum*. Their compositions were analyzed using different spectral techniques, such as ^1^H/^13^C NMR and HRMS, with the structure of **9l** being additionally confirmed through X-ray diffraction. The antifungal evaluation showed that, among all the target *β*-carboline analogues, compounds **9n** and **9o** exhibited more promising and broad-spectrum antifungal activity than the commercial pesticide hymexazol. Several intriguing findings regarding structure–activity relationships (SARs) were examined. In addition, the cytotoxicity test showed that these acylhydrazone *β*-carboline analogues with C1 substitutions exhibit a preference for fungi, with minimal harm to healthy cells (LO2). The reported findings provide insights into the development of *β*-carboline analogues as new potential antifungal agents.

## 1. Introduction

Plant pathogenic fungi have long been considered one of the main causative agents of plant diseases. They can infect any tissue at different stages of plant growth, leading to serious declines in the quality and yield of agricultural products. Several fungi-caused plant diseases have become major problems worldwide, including those caused by *Valsa mali*, *Fusarium graminearum*, *Fusarium oxysporum*, and *Fusarium solani* [1]. Plant diseases have a significant impact on the quality and quantity of crops, with fungal diseases exhibiting a wide range of severity and diversity [2]. Although traditional chemical fungicides are still the main control practice at present, the long-term use and misuse of chemical pesticides has led to the growing resistance of crop disease-causing agents, environmental pollution, and threats to human health [3,4]. Therefore, the development of new and green pesticides that are highly active, highly selective, low in toxicity, and environmentally friendly is urgently needed.

Natural bioactive products have played an important role in the discovery of pesticides and pharmaceuticals. For example, *β*-carboline, derived from the seeds of *Peganum harmala* L. in 1841 [5], and its analogues provide the core structure for a number of pharmaceutical and agrochemical activities, including anticancer [6], antibacterial [7], antiviral [8,9], and antifungal activities [10,11,12]. Earlier studies by Song et al. [13] and Hou et al. [14] showed that various *β*-carboline-2-ium salts and their derivatives displayed effective antiviral and antifungal properties. A variety of synthetic *β*-carboline analogues have recently been patented as fungicides for the treatment of plant diseases. Of medicinal relevance, harmane [15] and harmine [16] are reported to be potent Candida albicans inhibitors, and harmine in binary combinations with other carboline analogues strongly inhibits Aspergillus niger. Harmaline significantly inhibits Candida rugosa lipase in vitro as a competitive inhibitor according to in silico (docking) studies [17]. These and other examples attest to the potential of β-carboline analogues of natural or synthetic origin as antifungal agents for medicinal and other applications.

In recent years, the use of natural products to develop novel green pesticides has been favored by a large number of researchers [18,19,20,21,22]. For example, acylhydrazone moieties, the structure of which is -CONH–N=CH–, have received a great deal of attention as potential drug and pesticide formulation components [23,24]. Many commercially well-known fungicides include hydrazone moieties, such as Ferimzone [25] and Benquinox [26]. To the best of our knowledge, although many *β*-carboline derivatives have been reported, the introduction of acylhydrazone moieties on the C-1 position of *β*-carboline has not been reported. A set of new C1-substituted acylhydrazone *β*-carboline compounds were created and evaluated for their antifungal and cytotoxic effects as described above (Figure 1). These results are anticipated to facilitate the creation of novel antifungal medications based on *β*-carboline analogues.

## 2. Results and Discussion

### 2.1. Synthesis of C1-Substituted Acylhydrazone β-Carboline Analogues ***9a***–***o*** and ***10a***–***o***

Figure 1 illustrates the synthetic pathways for the C1-substituted acylhydrazone *β*-carboline analogues **9a**–**o** and **10a**–**o**. Tryptamine or 5-methoxytryptamine was first reacted with methyl 4-formylbenzoate in the presence of TFA in dichloromethane at room temperature to form tetrahydro-*β*-carboline 3/4. Subsequently, the tetrahydro-*β*-carboline 3/4 was reduced under Pd/C to give the C1-substituted *β*-carboline 5/6. The intermediate 5/6 was then reacted with hydrazine hydrate to yield a hydrazide derivative of *β*-carboline 7/8. Ultimately, the acylhydrazone *β*-carboline analogues **9a**–**o** and **10a**–**o** were synthesized by reacting compound 7/8 with different benzaldehyde substitutions to achieve the desired C1-substituted targets. The structures of compounds **9a**–**o** and **10a**–**o** were determined through a range of spectral analyses, such as ^1^H NMR, ^13^C NMR, and HRMS. More importantly, the structure of compound **9l** was further identified through X-ray diffraction (Figure 2), and its deposited CCDC number is 2195088.

### 2.2. Antifungal Activity and Structure–Activity Relationships (SARs)

A preliminary assessment was conducted to determine the antifungal properties of the C1-substituted acylhydrazone *β*-carboline analogues **9a**–**o** and **10a**–**o** against *V. mali*, *F. graminearum*, *F. oxysporum*, and *F. solani* using the mycelial growth rate method [27]. As displayed in Table 1, among the *β*-carboline analogues **9a**–**o**, compounds **9a**, **9n**, and **9o** showed greater potency of antifungal activity against *V. mali* than the commercial pesticide hymexazol; compounds **9h**, **9n**, and **9o** exhibited good antifungal effect against *F. solani* with all the inhibition rates exceeding 52%; compounds **9e**, **9j**, **9n**, and **9o** displayed potent antifungal effects against *F. oxysporum*; and finally, compounds **9b**, **9d**, **9h**, **9j**, **9n**, and **9o** exhibited promising antifungal activities against *F. graminearum*, with the inhibition rates all greater than 50%. To our delight, we found that compounds **9n** and **9o** exhibited more promising and broad-spectrum antifungal activity than hymexazol. In particular, the inhibition rates of compound 9n against the four fungi were 54.9%, 64.5%, 63.0%, and 67.0%, respectively, significantly higher than those of hymexazol. Of the *β*-carboline analogues **10a**–**o**, compounds **10d**, **10j**, **10n**, and **10o** exhibited strong antifungal activities against *F. oxysporum*, *F. graminearum*, and *F. solani*, with the inhibition rates exceeding 50%. Among them, the antibacterial activity of compound **10n** against *F. oxysporum* and *F. graminearum* was stronger than that of hymexazol. The experimental results were consistent with the reported results that β-carboline analogues have good antifungal effects [14,28].

Furthermore, the SARs yielded some intriguing findings. When the R1 group was an electron-donating group, we observed that the corresponding target analogue had lower antifungal activity, when compared to analogues with halogenated R1 groups (Figure 3A) (e.g., against *V. mali*, **9k** (R^1^ = 2-OCH_3_), **9l** (R^1^ = 4-OCH_3_), and **9m** (R^1^ = 4-CH_3_) vs. **9b** (R^1^ = 2-F), **9g** (R^1^ = 4-Cl), and **9j** (R^1^ = 4-Br). Additionally, the antifungal activities of target *β*-carboline analogues with an R group of -H (**9b**–**o**) were better than those with an R group of -OCH_3_ (**10b**–**o**) (Figure 3B). For example, the inhibition rates of **9b**–**o** (R = -H) against *V. mali* were 37.1%, 54.9%, and 52.1%, respectively. In contrast, the inhibition rates of **10m**–**o** (R = -OCH_3_) against *V. mali* were 35.0%, 48.5%, and 47.7%, respectively.

Furthermore, encouraged by the inhibition rates of **9n** and **9o** against the four fungi, their EC_50_ values against the four fungi were further studied. Table 2 shows that compound **9n** exhibited EC_50_ values of 27.8, 16.8, 16.2, and 16.1 μg/mL against *V. mali*, *F. solani*, *F. oxysporum*, and *F. graminearum*, while compound **9o** had EC_50_ values of 35.4, 21.5, 24.9, and 17.4 μg/mL, respectively, against the same fungi. In contrast, the EC_50_ values of the positive control hymexazol against these four fungi were 36.1, 31.9, 38.6, and 36.2 μg/mL, respectively. Taken together, both compounds **9n** and **9o** have greater potency and a broader spectrum of antifungal activity than hymexazol.

### 2.3. Cytotoxicity Assay

A good pesticide should have high selectivity, low toxicity to non-target organisms, and a good biosafety profile [19]. Therefore, the two potent compounds **9n** and **9o** were further screened for their toxicities to normal cell lines (LO2) using a CCK-8 assay [29]. A high cell viability of LO2 was maintained, even under treatment with 50 μg/mL of **9n** and **9o**. In contrast, the cell viability of LO2 was more strongly affected by hymexazol (Figure 4). These findings indicate that the *β*-carboline derivatives **9n** and **9o** exhibited a degree of specificity and resulted in minimal harm to healthy cells (LO2).

## 3. Experiment

### 3.1. Chemical Materials and Instruments

Aladdin Biotech Co (Hanzhong, China) supplied 5-methoxytryptamine and tryptamine, while other reagents were of analytical grade and did not require additional purification. The ^1^H/^13^C NMR profiles of each of the acylhydrazone *β*-carboline analogues with C1 substitution were analyzed with a 600 MHz Bruker Avance spectrometer from Bruker in Germany. The high-resolution mass spectra (HRMS) of all the target C1-substituted acylhydrazone *β*-carboline analogues were determined using an LTQ-FT Ultra instrument (Thermo Corporation, Waltham, MA, USA). The absorbance of the cell suspensions was determined at 450 nm using a microplate reader from BioTek in the Winooski, VT, United States.

### 3.2. Synthesis of Intermediates ***3*** and ***4***

To a solution of tryptamine (**1**, 2.0 mmol, 320.4 mg)/5-methoxytryptamine (**2**, 2.0 mmol, 386.5 mg) and methyl 4-formylbenzoate (2.2 mmol, 361.2 mg) in anhydrous CH_2_Cl_2_, trifluoroacetic anhydride (TFA, 2.6 mmol, 193 μL) was added slowly in a dropwise manner under an ice bath. The mixture was agitated for 24 h at ambient temperature and then filtered, and the resulting filtrate was dried to yield a 3/4 intermediate.

### 3.3. Synthesis of Intermediates ***5*** and ***6***

Compound **5**/**6** was obtained by dissolving intermediate **3** (1.0 mmol, 306.4 mg) or **4** (1.0 mmol, 336.4 mg) in 10 mL of toluene, adding 10% Pd/C (0.1 mmol), refluxing the reaction for approximately 48 h, filtering out the excess Pd/C after completion, and concentrating and purifying the filtrate using silica gel column chromatography (CC) with petroleum ether and EtOAc (*v*/*v* = 5:1) as the eluent.

### 3.4. Synthesis of Intermediates ***7*** and ***8***

A mixture of intermediate **5** (1.0 mmol, 302.4 mg)/**6** (1.0 mmol, 332.4 mg) and hydrazine hydrate (5 mmol, 250.3 mg) was refluxed in MeOH for about 24 h. Following the completion of the reaction, the solution was concentrated, and the remaining material was purified using silica gel column chromatography to obtain compound **7**/**8**.

### 3.5. Synthesis of C1-Substituted Acylhydrazone β-Carboline Analogues ***9a**–**o*** and ***10a**–**o***

Intermediate **7** (0.5 mmol, 151.2 mg) and Intermediate **8** (0.5 mmol, 166.2 mg) along with benzaldehyde (0.55 mmol) were dissolved in ethanol. Then, 100 μL of acetic acid was added to the mixture, which was then refluxed for 12 h. Once the reaction was finished, the solution was concentrated and separated using preparative thin layer chromatography (PTLC) to yield C1-substituted acylhydrazone *β*-carboline analogues **9a**–**o** and **10a**–**o**. The spectral information for the selected compounds **9a**, **9b**, **10a**, and **10b** is provided below; additional compounds **9c**–**o** and **10c**–**o** are detailed in the Appendix A.

**9a**: white solid; yield: 30%; ^1^H NMR (400 MHz, CDCl_3_) δ: 10.83 (s, 1H, H-9), 8.27 (s, 2H, -Ph), 7.93–7.89 (m, 5H, -Ph), 7.73 (d, *J* = 7.8 Hz, 1H, H-5), 7.49–7.40 (m, 3H), 7.29–7.26 (m, 1H), 7.12 (s, 3H), 7.03 (m, 1H); 13C NMR (100 MHz, CDCl_3_) δ: 163.3, 148.2, 141.7, 141.3, 141.2, 138.4, 134.3, 133.4, 133.0, 130.0, 129.6, 128.6, 128.4, 128.1, 127.2, 121.3, 120.9, 119.5, 114.1, 112.4; MS (ESI) *m*/*z* calcd for C_25_H_18_N_4_O ([M + H]+), 391.1553; found, 391.1540.

**9b**: white solid, yield: 53%; ^1^H NMR (600 MHz, DMSO-*d*_6_) δ: 12.20 (s, 1H, H-9), 11.66 (s, 1H), 8.81 (s, 1H), 8.51 (d, *J* = 4.8 Hz, 1H), 8.29 (d, *J* = 7.8 Hz, 1H), 8.21–8.19 (m, 5H, -Ph), 8.01–7.99 (m, 1H), 7.67 (d, *J* = 7.8 Hz, 1H), 7.59–7.57 (m, 1H), 7.53–7.51 (m, 1H), 7.35–7.28 (m, 3H); ^13^C NMR (150 MHz, DMSO-*d*_6_) δ: 161.1, 141.0, 140.6, 140.4, 140.1, 137.9, 132.6, 132.1, 131.5 (*J_CF_* = 9.0 Hz), 131.5, 131.4, 128.9, 127.9, 127.8, 127.1, 124.4 (*J_CF_* = 1.5 Hz), 124.4, 124.4, 121.1, 120.2, 119.1, 115.5 (*J_CF_* = 21.0 Hz), 115.5, 115.4, 113.9, 111.8; MS (ESI) *m/z* calcd for C_25_H_17_FN_4_O ([M + H]+), 409.1459; found, 409.1469.

**10a**: white solid, yield: 85%; ^1^H NMR (600 MHz, DMSO-*d*_6_) δ: 12.04 (s, 1H, H-9), 11.47 (s, 1H), 8.55 (s, 1H), 8.47 (d, *J* = 5.4 Hz, 1H), 8.19–8.16 (m, 5H, -Ph), 7.84 (d, *J* = 2.4 Hz, 1H), 7.78 (d, *J* = 7.2 Hz, 2H), 7.57 (d, *J* = 9.0 Hz, 1H), 7.50–7.48 (m, 3H), 7.23–7.22 (m, 1H), 3.89 (s, 3H, -OCH_3_); ^13^C NMR (150 MHz, DMSO-*d*_6_) δ: 162.1, 153.0, 147.4, 140.9, 140.4, 137.3, 135.4, 13.8, 133.1, 128.8, 128.3, 127.7, 127.4, 126.5, 120.5, 117.8, 113.9, 112.7, 102.8, 55.0; MS (ESI) *m*/*z* calcd for C_26_H_20_N_4_O_2_ ([M + H]+), 421.1659; found, 421.1667.

**10b**: white solid, yield: 51%; ^1^H NMR (600 MHz, DMSO-*d*_6_) δ: 11.18 (s, 1H, H-9), 11.47 (s, 1H), 8.80 (s,1H), 8.47 (d, *J* = 5.4 Hz, 1H), 8.20–8.16 (m, 5H, -Ph), 8.00 (t, *J* = 7.2 Hz, 1H), 7.84 (d, *J* = 2.4 Hz, 1H), 7.58–7.51 (m, 2H), 7.35–7.32 (m, 2H), 7.22 (dd, *J* = 9.0 Hz, 2.4 Hz, 1H), 3.89 (s, 3H, -OCH_3_); 13C NMR (150 MHz, DMSO-*d*_6_) δ: 163.2, 160.5, 154.1, 142.1, 141.5, 141.1, 138.4, 136.5, 134.2, 133.2, 132.6, 129.9, 128.7 (*J_CF_* = 46.5 Hz), 128.9, 128.6, 126.8, 125.5 (*J_CF_* = 3.0 Hz), 125.5, 125.5, 121.6, 118.9, 116.5 (*J_CF_* = 15.0 Hz), 116.6, 116.5, 115.0, 113.8, 103.9, 56.1; MS (ESI) *m*/*z* calcd for C_26_H_19_FN_4_O_2_ ([M + H]+), 439.1565; found, 439.1576.

### 3.6. Biological Evaluation

#### 3.6.1. Antifungal Activity

The effectiveness of the *β*-carboline derivatives **9a**–**o** and **10a**–**o** in inhibiting the growth of four plant-damaging fungi (*F. solani*, *F. graminearum*, *V. mali*, and *F. oxysporum*) was assessed using the mycelial growth rate technique as described in earlier studies [24]. To prepare the PDA culture medium, potato dextrose agar was used, followed by 30 min of sterilization at 115 °C. Next, 5 mm diameter mycelial disks were excised from each fungus using a sterile cork borer and then cultured in PDA at 28 ± 1 °C for around 5 days in order to obtain the mycelia for antifungal testing. Subsequently, PDA Petri dishes were inoculated with mycelial discs. Acetone was used as a solvent to dissolve the hymexazol, as well as the *β*-carboline analogues **9a**–**o** and **10a**–**o**, before being combined with the culture medium at a concentration of 50 g/mL. Following that, sterilized Petri dishes were filled with the mixed medium. Following inoculation, the cells were incubated at a temperature of 28 ± 2 °C for a period of 3 days. Hymexazol, a commercial fungicide, served as the positive control, while acetone with PDA was used as the blank control (CK). The colony’s radial growth was measured with a vernier caliper after being subjected to each treatment three times.

The determination of the 50% effective concentration (EC_50_) was carried out using the same experimental method described above [2]. Gradients of 80, 40, 20, 10, and 5 μg/mL were set for the determination of the compounds, and three replicates were set for each tested concentration, and the results were statistically analyzed after all the inhibition data were obtained to derive the antifungal regression equations and EC_50_ values.

#### 3.6.2. Cytotoxicity Assay

The toxicity of the compounds **9n**, **9o**, and hymexazol on the LO2 cells was assessed using the cell counting kit-8 (CCK-8) assay, as described in the literature [29,30]. The LO2 cells were incubated at 37 °C for a day, followed by the addition of 100 μL of new medium with varying levels of **9n**, **9o**, and orhymexazol to replace the old medium. Only fresh medium was used for the negative controls. Fresh medium was used after the used medium was discarded after 24 h of incubation. Following that, the cells were cultured in a new solution with 5% CCK-8 for an additional 4 h at 37 ° C. Subsequently, the absorbance at 450 nm was measured using a microplate reader.

#### 3.6.3. Data Analysis

To calculate the inhibition rate of fungi by compounds **9a**–**o** and **10a**–**o**, the following formula was used: inhibition rate (%) = (B − I) × 100/(B − 5 mm), where B represents the fungal growth diameter of the blank group, and I represents the fungal growth diameter of the compound group. The percentage of the cell survival rate was determined according to the following formula: cell survival rate (%) = (OD _experimental_ − OD _blank_)/(OD_negative control_ − OD _blank_) × 100%.The mean ± SD was calculated three times for each group. The SPSS 21.0 software was used for statistical analysis.

## 4. Conclusions

To summarize, we developed a range of C1-replaced acylhydrazone *β*-carboline derivatives **9a**–**o** and **10a**–**o** and tested their effectiveness against *V. mali*, *F. solani*, *F. oxysporum*, and *F. graminearum*. The structures of each of the target *β*-carboline analogues **9a**–**o** and **10a**–**o** were determined through a range of spectral analyses, including ^1^H/^13^C NMR, and HRMS. Additionally, the structure of **9l** was confirmed using X-ray diffraction. The antifungal evaluation showed that, among all the target *β*-carboline analogues, **9n** and **9o** exhibited more promising and broad-spectrum antifungal activity than the commercial pesticide hymexazol. The SARs revealed that *β*-carboline analogues with an R1 group as the halogenated group and an R group as -H had more promising antifungal activity than those with an R1 group as the electron-donating group and an R group as -OCH_3_. In addition, the cytotoxicity test demonstrated that these acylhydrazone *β*-carboline analogues with C1 substitutions exhibit a certain preference for fungi, with minimal harm to healthy cells (LO2). Based on the aforementioned findings, these acylhydrazone *β*-carboline analogues with C1 substitutions have the potential to be considered for the creation of novel antifungal medications.

## Data Availability

The data are included in the article and Appendix A.

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
