# Peer review of "Synthesis and Biological Activities of C1-Substituted Acylhydrazone β-Carboline Analogues as Antifungal Candidates"

_molecules, 2024, doi:10.3390/molecules29153569_

Round 1

Reviewer 1 Report

Comments and Suggestions for Authors

The authors of the manuscript 'Synthesis and biological activities of C1-substituted acylhydrazone β-carboline analogues as antifungal candidates' synthesized and spectroscopically characterized (1H, 13C NMR, HRMS) a large series of new organic compounds from the acylhydrazone β-carboline group. The authors also confirmed the receipt and structure of the assumed substances by X-ray structural analysis of the selected derivative. The authors synthesized compounds to obtain substances with antifungal activity. The compounds obtained were mostly weaker or comparable to the reference compound used. The two obtained compounds turned out to be more active than hymexazole.

In my opinion, the manuscript is interesting and well-written. With some revisions, it may be considered for publication on Molecules.

First of all, the authors should complete the supplement with HR MS spectra.

Did the authors observe scattering of carbon signals combined with fluorine in 13C NMR spectra? In my opinion this should be visible. In this case, coupling constants must be provided.

The authors should also revise Scheme 1. It is about explaining the structure of R1 substituents, according to Scheme 1, a benzene ring with an R1 substituent is connected to the NH-N=C- fragment, in the explanation of R1 a=Ph, are you sure? It should be a=H, b=2-Br etc (at least this is what follows from the structure of substance 9l shown in Figure 2.)

The authors should also check the manuscript for editing (compound numbers are in bold, change 13C to 13C, format references according to MDPI requirements).

Author Response

The authors of the manuscript 'Synthesis and biological activities of C1-substituted acylhydrazone β-carboline analogues as antifungal candidates' synthesized and spectroscopically characterized (1H, 13C NMR, HRMS) a large series of new organic compounds from the acylhydrazone β-carboline group. The authors also confirmed the receipt and structure of the assumed substances by X-ray structural analysis of the selected derivative. The authors synthesized compounds to obtain substances with antifungal activity. The compounds obtained were mostly weaker or comparable to the reference compound used. The two obtained compounds turned out to be more active than hymexazole.

In my opinion, the manuscript is interesting and well-written. With some revisions, it may be considered for publication on Molecules.

Response: We really appreciate your positive comments. We have supplemented some data and revised the manuscript according to your constructive suggestions.

Special comments:

     1. First of all, the authors should complete the supplement with HR MS spectra.

Response: Special thanks to the reviewer for your advice. We have supplemented the HRMS spectra in supplementary material (Figure S59- S88) as you suggested.

  1. Did the authors observe scattering of carbon signals combined with fluorine in 13C NMR spectra? In my opinion this should be visible. In this case, coupling constants must be provided.

Response: Many thanks for the reviewer’s insightful comment and advice. We observe scattering of carbon signals combined with fluorine in 13C NMR spectra of several compounds, we have added the coupling constants as the reviewer suggested and the results were presented in supporting information. ( Supplementary Material Page: 2,3,8-10 )

  1. The authors should also revise Scheme 1. It is about explaining the structure of R1 substituents, according to Scheme 1, a benzene ring with an R1 substituent is connected to the NH-N=C- fragment, in the explanation of R1 a=Ph, are you sure? It should be a=H, b=2-Br etc (at least this is what follows from the structure of substance 9l shown in Figure 2.)

Response: Thank you for your careful reading. We are very sorry for this clerical error, and we have revised it in the revised manuscript.

Page 9: Furthermore, SARs yielded some intriguing findings. When the R1 group was an electron-donating group, we observed that the corresponding target analogue had lower antifungal activity, when compared to analogues with halogenated R1 groups (Figure 3A) (e.g., against V. mali, 9k (R1 = 2-OCH3), 9l (R1 = 4-OCH3), and 9m (R1 = 4-CH3) vs 9b (R1 = 2-F), 9g (R1 = 4-Cl), and 9j (R1 = 4-Br). Additionally, the antifungal activities of target β-carboline analogues with an R group of -H (9b−o) were better than those with an R group of -OCH3 (10b−o) (Figure 3B). For example, the inhibition rates of 9b−o (R = -H) against V. mali were 37.1%, 54.9%, and 52.1%, respectively. In contrast, the inhibition rates of 10m−o (R = -OCH3) against V. mali were 35.0%, 48.5%, and 47.7%, respectively.

Scheme 1. Synthesis of C1-substituted acylhydrazone β-carboline analogues.

  1. The authors should also check the manuscript for editing (compound numbers are in bold, change 13C to 13C, format references according to MDPI requirements).

Response: Thank you so much for your critical reading of our manuscript. Now we have carefully checked the whole manuscript and corrected these errors in the revised manuscript. In addition, we have corrected all the references in the revised manuscript.

Reviewer 2 Report

Comments and Suggestions for Authors

Comments on the Quality of English Language

Author Response

Manuscript title: Synthesis and biological activities of C1-substituted acylhydrazone β-carboline analogues as antifungal candidates The presented manuscript describes the synthesis of C1-substituted acylhydrazone β-carboline analogues and the examination of their antifungal potential and citotoxicity. This study seems well designed and deals with an interesting and contemporary problem, but has some flaws that should be revised to enhance the overall quality of the manuscript.

Response: Many thanks to the reviewer for your careful review and constructive comments. All those comments are valuable and helpful for revising and improving our paper, as well as the important guidance to our future researches. The main corrections in the paper and the responses to the reviewer's comments are as follow.

  1. Latin names of organisms should be italicized, please correct where needed.

Response: Thank you so much for your critical reading of our manuscript. We have corrected the format of latin names of organisms in the revised manuscript.

  1. “More recent years” – In recent years?

Response: Thank you very much for your good suggestion. We have revised it in the revised manuscript according to your advice.

  1. There is no need to describe results in the Introduction section.

Response: Many thanks for the reviewer’s insightful comment. We have deleted this section in the revised manuscript.

  1. In Figure 1, maybe mark the β-carboline moiety on the first two compounds with a different color, and for the compounds in this work, keep only the general structure with both moieties marked. The Scheme 1 describes in detail the synthesis steps and it is clear there what the starting compounds are.

Response: Special thanks to the reviewer for your advice. We have revised the Figure 1 according to your advice, and mark the β-carboline moiety with blue.

  1. Please correct throughout the manuscript, it is Valsa mali, not Valsa mail!

Response: Thank you very much for your reminder. We have corrected it already in the revised manuscript.

  1. There are 30 compounds in total, but 28 are shown in Figure 3!

Response: Many thanks for the reviewer's question. The discussion here is based on the conformational analysis of the substituent R1 on the benzene ring and discusses the change in activity produced by the substituent group on R1, meaning that the comparison is done on the basis of compounds 9a (R1=H) and 10a (R1=H), but not including these two compounds.

  1. The discussion could be enhanced by incorporating additional pertinent papers. The current literature review is also limited, citing only a few relevant research papers. Consider supplementing the discussion with the latest research findings related to plant pathogens investigates for a more comprehensive analysis.

Response: Many thanks to the reviewer for your constructive comments. We have revised it in the manuscript according to your advice.

Page 1: Plant pathogenic fungi have long been considered one of the main causative agents of plant diseases. They can infect any tissue at different stages of plant growth, leading to serious declines in the quality and yield of agricultural products.

Page 2: A variety of synthetic β-carboline analogues analogs have recently been patented as fungicides for the treatment of plant diseases. Of medicinal relevance, harmane [15] and harmine [16] are reported to be potent Candida albicans inhibitors and harmine in binary combinations with other carboline analogues strongly inhibits Aspergillus niger. Harmaline significantly inhibits Candida rugosa lipase in vitro as a competitive inhibitor according to in silico (docking) studies [17]. These and other examples attest to the potential of β-carboline analogues of natural or synthetic origin as antifungal agents for medicinal and other applications.

Page 5: Among them, the antibacterial activity of compound 10n against F. oxysporum and F. graminearum was stronger than that of hymexazol. The experimental results were consistent with the reported results that β-carboline analogues have good antifungal effects [29-31].

References

[15] Reza V. R. M. and Abbas H., Cytotoxicity and antimicrobial activity of harman alkaloids. Journal of Pharmacology and Toxicology. 2007, 7, 677–680.

[16] Nenaah G. Antibacterial and antifungal activities of (beta)-carboline alkaloids of Peganum harmala (L) seeds and their combination effects. Fitoterapia. 2010, 81(7), 779-782.

[17] Benarous K, Bombarda I, Iriepa I, et al. Harmaline and hispidin from Peganum harmala and Inonotus hispidus with binding affinity to Candida rugosa lipase: In silico and in vitro studies. Bioorganic Chemistry. 2015, 62, 1-7.

[29] Dai JK, Dan WJ, Wan JB. Natural and synthetic β-carboline as a privileged antifungal scaffolds. European Journal of Medicinal Chemistry. 2022, 229, 114057.

[30] Sheng T, Yu C, Wang Y, et al. Chromatography-Free Synthesis of β-Carboline 1-Hydrazides and an Investigation of the Mechanism of Their Bioactivity: The Discovery of β-Carbolines as Promising Antifungal and Antibacterial Candidates. Journal of Medicinal Chemistry. 2023, 66(13), 9040-9056.

[31] Hou Z, Zhu LF, Yu XC, Sun MQ, Miao F, Zhou L. Design, Synthesis, and Structure--Activity Relationship of New 2-Aryl-3,4-dihydro-β-carbolin-2-ium Salts as Antifungal Agents. Journal of Agricultural and Food Chemistry. 2016, 64(14), 2847-2854.

  1. In the Experimental section, there is no mention of how the EC50 values for compounds 9n, 9o and hymexazol were determined, please add. Furthermore, add the graphs for EC50 in the Supplementary, since the regression equation is given in the manuscript.

Response: Many thanks to the reviewer for your careful review and constructive comments. We have added a description of the EC50 assay as the reviewer suggested. Page 11: the determination of 50% effective concentration (EC50) was carried out using the same experimental method described above [30]. Gradients of 80, 40, 20, 10, and 5 μg/mL were set for the determination of compounds, and three replicates were set for each tested concentration, and the results were statistically analyzed after all the inhibitions obtained to derive antifungal regression equations and EC50 values. In addition, we have supplemented the graphs for EC50 in the revised supporting information. (Figure S89-S91)

References

[30] Sheng T, Yu C, Wang Y, et al. Chromatography-Free Synthesis of β-Carboline 1-Hydrazides and an Investigation of the Mechanism of Their Bioactivity: The Discovery of β-Carbolines as Promising Antifungal and Antibacterial Candidates. Journal of Medicinal Chemistry. 2023, 66(13), 9040-9056.

  1. Is there a valid reason for referencing (+)-nootkatone derivatives four times in this paper? Out of 27 references, 4 are about them, surely there are more appropriate ones. There are many papers describing different types of bioactive compounds tested for pesticidal activity.

Response: Thank you so much for your critical reading of our manuscript. We have corrected the references in the revised manuscript.

  1. Check the entire manuscript for typos, missing spaces and grammatical errors.

Response: Many thanks for the reviewer’s insightful advice. Now we have carefully checked the whole manuscript and corrected these errors in the revised manuscript.

  1. In Supplementary, check thoroughly data reported, proton numbers and signals do not match in some compounds. Furthermore, why are there no 13C NMR spectra for compounds 10d, 10f, and 10o?

Response: Thank you for your critical and careful reading. We apologize for our carelessness and now we have carefully checked the whole supporting information and modified some errors. In addition, we have supplemented the 13C NMR data and 13C NMR spectra of compound 10d in the revised supporting information. However our compounds 10f and 10o were used up after the antifungal experiments, but we were able to determine their structures based on the 1H NMR and HRMS data obtained.

Round 2

Reviewer 2 Report

Comments and Suggestions for Authors

The authors have addressed and corrected all the issues raised in the review. The manuscript has been improved, there may be some typos still but it can be resolved in the proofreading step.